# Converse flexoelectric two-dimensional MoS$_2$ actuator

Yeageun Lee [1,2,5], Hyung Jong Bae [1,5], Md Farhadul Haque[1], Keon-Hee Lim[1], Jin Myung Kim[2,3], Weilin Guan[2] & SungWoo Nam [2,4] ✉

Converse flexoelectric actuators offer significant potential for diverse applications due to their rapid response, resilience in extreme environments, and broad material compatibility. However, their development has been limited, primarily because flexoelectric effects are negligible in bulk materials. At the nanoscale, the converse flexoelectric effect becomes much more promising, as the electric field gradient scales quadratically with decreasing material thickness. Here, we report a converse flexoelectric actuator based on two-dimensional molybdenum disulfide. Under alternating current excitation near 20 kHz, the actuator exhibits resonant displacements up to ~45 nm, approximately two orders of magnitude larger than the thickness of the molybdenum disulfide active layer. This performance exceeds that of existing flexoelectric actuators by more than an order of magnitude when normalized by active layer thickness. Moreover, the actuator maintains strong flexoelectric responses under extreme conditions, including vacuum, cryogenic temperatures, and repeated cycling, highlighting the robustness and broad applicability of two-dimensional material-based converse flexoelectric systems.

With the growing demand for smaller systems and precise controls for nano/micro-robotics, nanoscale metrology and manipulating instruments, and aerospace applications, various types of nanoscale actuators have been actively explored. These nano-actuators would have to satisfy a number of requirements such as force, displacement, response speed, and repeatability. Among several types of nano-actuators including piezoelectric, electrostatic, electrostrictive, magenetostrictive, and thermal expansion-based actuators, piezoelectric actuators (piezo-actuators) are favorable for many applications due to their fast response, suitability for vacuum condition, nanoscale controllability, and compact size[1,2]. Nevertheless, several drawbacks also exist: i) only non-centrosymmetric crystals exhibit piezoelectric effect[3], ii) a majority of piezo-materials contain heavy metals such as lead which are not biocompatible[3,4], iii) the stroke of piezo-actuators is significantly low relative to the actuators' size[2,5,6], and iv) piezoelectric effect decreases sharply in cryogenic environment[7–9].

Flexoelectric (or converse flexoelectric) effect, a similar electromechanical effect to the piezoelectric effect, is an electric polarization under non-zero strain gradient (or a mechanical response under non-zero electric field (E-field) gradient). Flexoelectricity exists in all dielectrics including centrosymmetric materials, and consequently, flexoelectric actuators (flexo-actuators) are a promising candidate to overcome aforementioned first and second limitations of piezo-actuators. In addition, since the mechanical response from converse flexoelectricity increases quadratically as the material's thickness decreases, there is room for improving the relative stroke with extremely thin active layer[10–12]. Although there have been a few recent studies related to flexo-actuators, the active materials were in micro/millimeter scale, and thus the strokes were small relative to their size[13–16]. In this respect, two-dimensional (2D) materials can play a critical role as active materials for the flexo-actuators due to their atomically thinness and

[1]Department of Mechanical Science and Engineering, University of Illinois at Urbana Champaign, Urbana, IL, USA. [2]Department of Mechanical and Aerospace Engineering, University of California, Irvine, Irvine, CA, USA. [3]Department of Materials Science and Engineering, University of Illinois at Urbana-Champaign, Urbana, IL, USA. [4]Department of Materials Science and Engineering, University of California, Irvine, Irvine, CA, USA. [5]These authors contributed equally: Yeageun Lee, Hyung Jong Bae. ✉e-mail: sungwoo.nam@uci.edu

mechanical flexibility. Although the flexoelectric effect of 2D materials has been explored in both theoretical and experimental studies, most works have focused on materials-level behavior or computational modeling[12,17–19]. In contrast, an actuator driven by converse flexoelectric response of 2D materials has not been demonstrated to the best of our knowledge.

Here we show a flexo-actuator operated by converse flexoelectric effect leveraging 2D materials. E-field gradient is generated in a monolayer molybdenum disulfide ($MoS_2$) through asymmetric electrode design to actuate a 600 nm thick beam-type structure. In-plane strain and strain gradient generated by the $MoS_2$ active layer induce ~45 nm of dynamic out-of-plane actuation under 40 V peak-to-peak ($V_{pp}$) AC voltage. This actuation level is more than an order of magnitude higher than that of reported flexo-actuators when normalized by active layer thickness and applied voltage[13–15], and it is even comparable to piezo-actuators[20–30]. In addition, the actuation of our device can be controlled with nanoscale precision by modulating the applied voltage, since the actuation is linearly proportional to the applied voltage. Furthermore, our device shows robust performance up to $10^{10}$ cycles and maintains ~70% of its actuation under 10 K cryogenic condition in contrast to the conventional piezo-actuator maintaining only ~40% of its actuation under the same condition. Our results highlight the novelty of flexoelectricity in 2D materials, demonstrating substantial scaling advantages and strong potential for cryogenic device applications.

## Results

### 2D material-based flexo-actuator

Figure 1a shows the design of our flexo-actuator. Our device is composed of four layers: patterned 50 nm-Au top electrode, monolayer $MoS_2$ active layer, 500 nm-Parylene-C insulating/supporting layer, and 50 nm-Ag bottom electrode (see Methods for fabrication details). In order to generate a strong E-field gradient in the $MoS_2$ active layer, the top electrode was patterned to have a comb-like shape with 20 μm pitch (Fig. 1b), while a metal electrode without pattern was used as the bottom electrode. The multilayered structure was then transferred on the sample holder having ~1 mm gap in the middle to form a suspended beam structure (Fig. S1). Photoluminescence (PL) mapping showing strong and uniform PL intensity at 665 nm across our device demonstrated the quality and uniformity of the $MoS_2$ active layer (Fig. 1c)[31].

To characterize the mechanical actuation of our flexo-actuator device, a function generator, an amplifier, and our device were connected in series to apply a sinusoidal AC voltage to the device (Fig. 1d). When the AC voltage induced the deformation of the flexo-actuator, the resultant motion of our device was measured by a laser vibrometer as a form of vertical velocity. Both the applied voltage and vertical velocity signals were observed by an oscilloscope in real time (Fig. 1d and Fig. S2). The displacement ($s$) of the device was then calculated from the vertical velocity ($v$) and frequency ($f$) by $s = v/2\pi f$.

After setting up the testing environment, we began by measuring the displacement of the flexo-actuator device under different

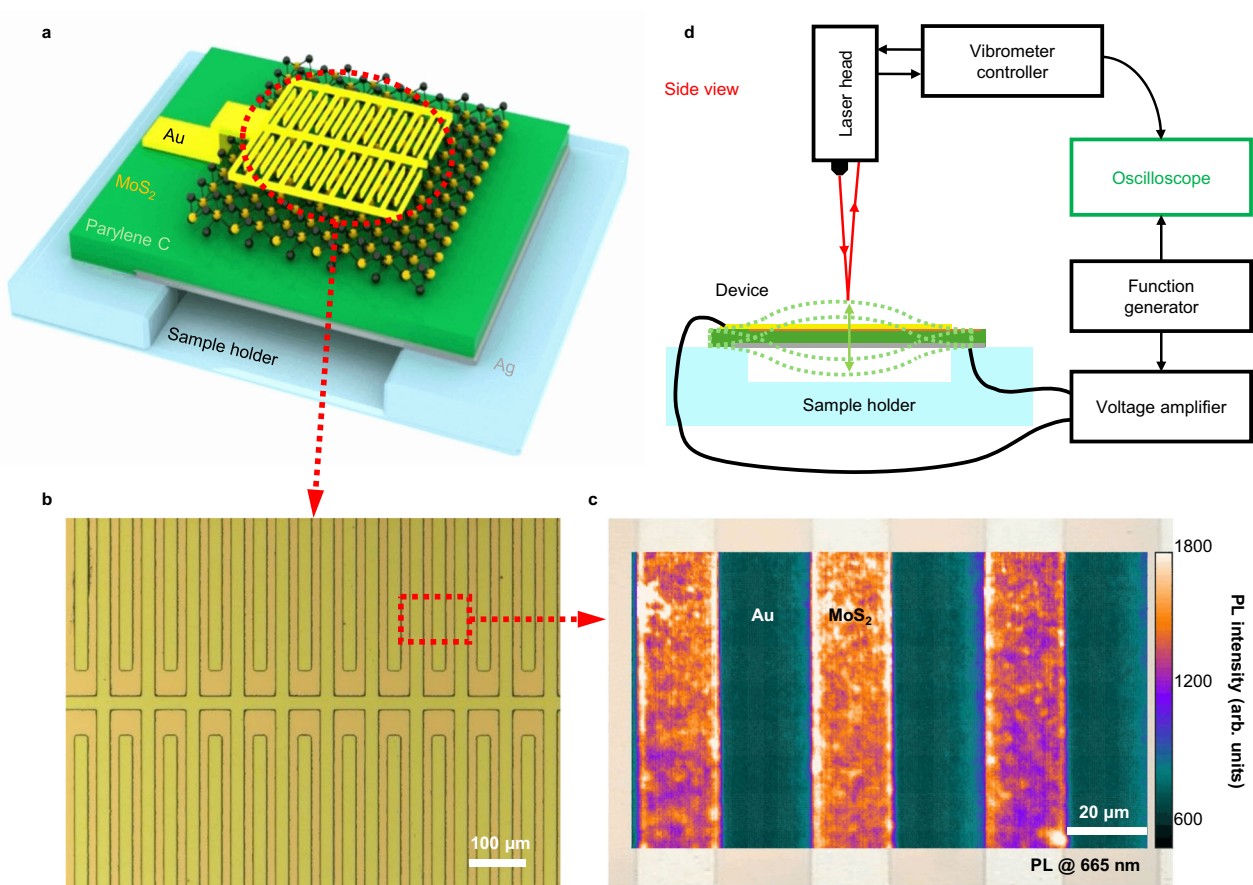

**Fig. 1 | Converse flexoelectric MoS₂ actuator. a** Schematic diagram of the converse flexoelectric $MoS_2$ actuator. The flexo-actuator has 4-layer: comb-shape Au top electrode, $MoS_2$ active layer, Parylene-C supporting/insulating layer, and Ag bottom electrode. **b** Optical microscope image of the Au/MoS₂/Parylene-C/Ag structure. The top Au electrode was patterned to have a uniform comb-shape structure. **c** Photoluminescence (PL) intensity mapping of the Au/MoS₂/Parylene-C/ Ag structure at 665 nm. Exposed $MoS_2$ area shows high PL intensity whereas the Au comb-shape electrode area shows low PL intensity. **d** Schematic diagram of the measurement setup. Amplified AC voltage was applied to the flexo-actuator through function generator and voltage amplifier. The actuation of the flexo-actuator was then observed by a laser vibrometer.

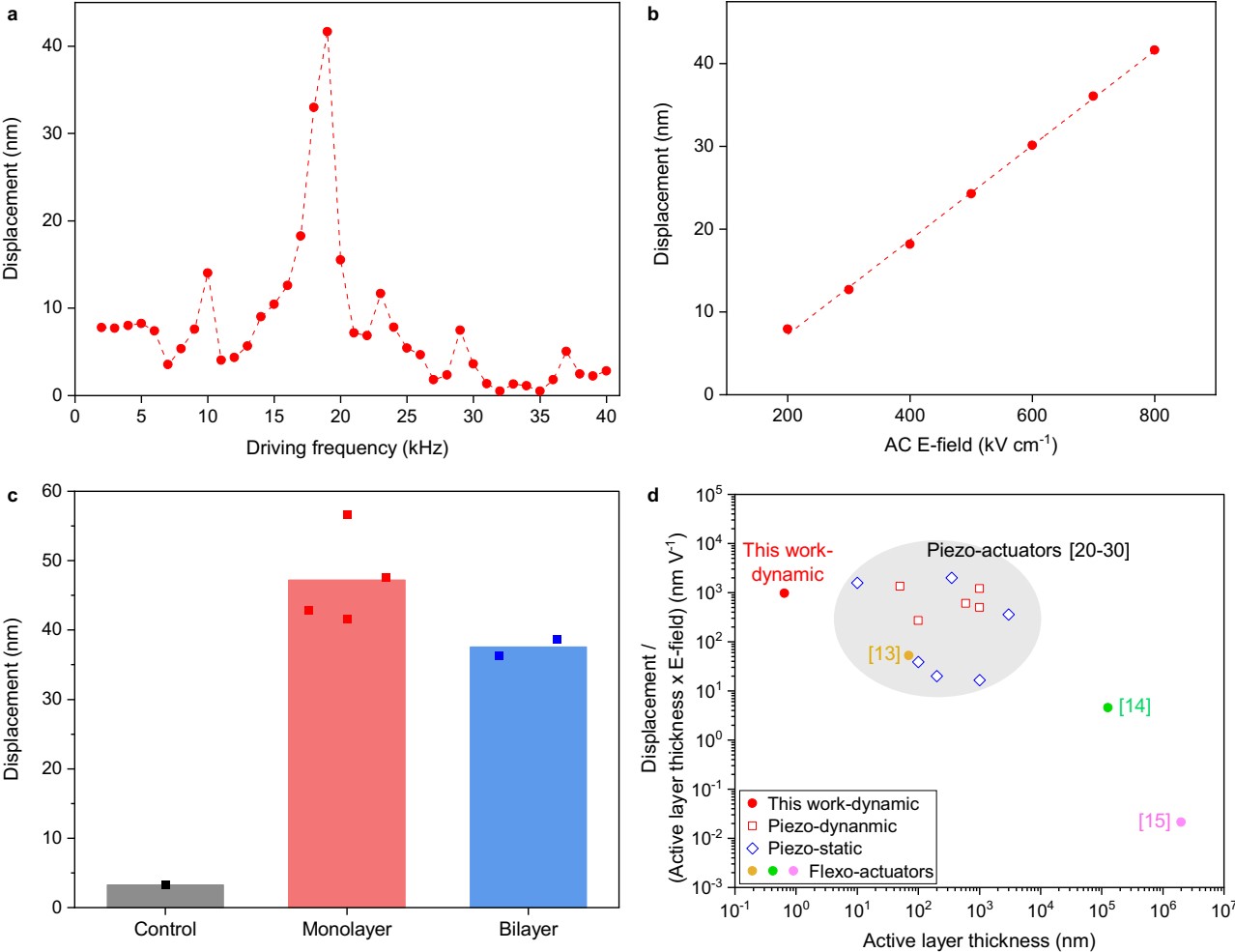

**Fig. 2 | Actuation performance of the converse flexoelectric MoS$_2$ actuator.**
**a** Dynamic displacement of the monolayer MoS$_2$ flexo-actuator at different AC frequency. The peak-to-peak voltage was maintained at 40 V. **b** Device displacement at resonant frequency (19 kHz) under different AC E-field. AC E-field was controlled by varying peak-to-peak voltage. **c** Resonant displacement of the three different devices under 40 V$_{pp}$: Devices without MoS$_2$ active layer (Control), Devices with monolayer MoS$_2$ active layer (Monolayer), and Devices with bilayer MoS$_2$ active layer (Bilayer). Bars and dots indicate the mean value and individual data points, respectively. **d** Performance comparison with other piezo- and flexo-actuators through the displacement normalized by active layer thickness and applied E-field. Refs. 13–15: flexo-actuators, refs. 20–24: dynamic piezo-actuators, and refs. 25–30: static piezo-actuators.

frequencies of 40 V$_{pp}$ AC voltage. This measurement provided a displacement profile and maximum displacement of the flexo-actuator device. As shown in Fig. 2a, our flexo-actuator device exhibited a sharp peak near 19 kHz and actuated up to 42 nm when 40 V$_{pp}$ of AC voltage was applied.

We next measured the displacement of the flexo-actuator device under various applied AC voltages (10 ~ 40 V$_{pp}$) while keeping the frequency of the applied AC voltage at the resonant frequency (19 kHz). This measurement was used to determine the relationship between the device displacement and the applied AC voltage. As shown in Fig. 2b, the displacement of our flexo-actuator device linearly increased as the applied AC E-field increased. This linearity demonstrates that the actuation of the flexo-actuator device can be controlled by simply adjusting the applied AC voltage. Additionally, the linear relationship supports that the actuation was governed by converse flexoelectricity, as the mechanical response of the converse flexoelectricity is linearly proportional to the applied E-field gradient[32].

To study the origin of our device actuation, we fabricated and characterized three different types of devices: i) devices without MoS$_2$ layer (control samples), ii) devices with monolayer MoS$_2$, and iii) devices with bilayer MoS$_2$. We observed linear AC E-field versus

displacement for all three types of devices with different displacement levels (Figure S3). Under 40 V of V$_{pp}$, the control samples showed less than 5 nm of actuation, whereas the devices with monolayer and bilayer MoS$_2$ exhibited 45 nm and 37 nm of average actuation, respectively (Fig. 2c). These results suggest important implications for the actuation mechanism. First, since the average displacement of the control devices is much smaller than that of monolayer and bilayer MoS$_2$ devices, the displacement can be attributed primarily to the MoS$_2$ active layer. Second, the similar level of displacement between monolayer and bilayer MoS$_2$ suggests that the main cause of actuation is likely due to converse flexoelectricity. It has been understood that only odd-layer MoS$_2$ exhibits in-plane piezoelectricity[33]. If in-plane piezoelectricity dominated the actuation, we should have observed much smaller displacement from bilayer MoS$_2$ because of diminishing in-plane piezoelectricity.

To put our results in perspective with other beam-type nano-/micro-piezoelectric and flexoelectric devices, we compared the actuation performance of our device with other reported values. As a performance factor, we used displacement normalized by active layer thickness times E-field, i.e., displacement / (active layer thickness × E-field). This factor indicates the degree of device deformation with unit thickness of active layer under the same E-field. The performance

factors of various actuators are plotted in Fig. 2d. Our device showed approximately $10^3$ nm/V of performance factor which is more than an order of magnitude larger than that of state-of-the-art flexoelectric devices (refs. 13–15 in Fig. 2d). Furthermore, our device also exhibited comparable performance factor compared to the beam type piezo-actuators having similar actuation mechanism with our device (refs. 20–24 in Fig. 2d for dynamic piezo-actuators and refs. 25–30 for static piezo-actuators)[20–30].

## Computational study to verify the actuation mechanism

For further investigation on the actuation mechanism, we conducted computational study on our flexo-actuator. Since the maximum vertical displacement of the flexo-actuator (>40 nm) is approximately two orders of magnitude larger than the thickness of monolayer MoS$_2$ (~0.65 nm), a vertical expansion of MoS$_2$ layer due to vertical stress ($\sigma_{33}$) would be an implausible source of the actuation. In contrast, in-plane stress ($\sigma_{11}$) generated in MoS$_2$ layer can induce a beam bending (Fig. S4). If we assume both piezoelectric and flexoelectric effects are contributing mechanisms, in-plane stresses can be derived from Eq. (1), where $\varepsilon$, $\mu$ and $E$ are converse piezoelectric coefficient, converse flexoelectric coefficient, and E-field, respectively[32]. Subscripts 1, 2, and 3 indicate directions of beam's length, width, and height, respectively.

$$\sigma_{11} = \varepsilon_{11}E_1 + \varepsilon_{12}E_2 + \varepsilon_{13}E_3 + \mu_{1111}\partial E_1/\partial x_1 + \mu_{1122}\partial E_2/\partial x_2 + \mu_{1133}\partial E_3/\partial x_3$$
$$+ \mu_{1112}\partial E_1/\partial x_2 + \mu_{1113}\partial E_1/\partial x_3 + \mu_{1123}\partial E_2/\partial x_3 + \mu_{1121}\partial E_2/\partial x_1$$
$$+ \mu_{1131}\partial E_3/\partial x_1 + \mu_{1132}\partial E_3/\partial x_2$$

$$(1)$$

For monolayer MoS$_2$ (2H phase, D$_{3h}$ point group), Eq. (1) can be reduced to Eq. (2) [18,34].

$$\sigma_{11} = \varepsilon_{11}E_1 + \varepsilon_{12}E_2 + \varepsilon_{13}E_3 + \mu_{1111}\partial E_1/\partial x_1 + \mu_{1122}\partial E_2/\partial x_2 + \mu_{1133}\partial E_3\partial x_3$$

$$(2)$$

In our case, since there is no E-field and E-field gradient in the 2-direction, the second ($\varepsilon_{12}E_2$) and fifth ($\mu_{1122}\partial E_2/\partial x_2$) terms in Eq. (2) become zero. In addition, monolayer MoS$_2$ exhibits zero $\varepsilon_{13}$ due to inversion symmetry in the out-of-plane direction, and thus the third term ($\varepsilon_{13}E_3$) also becomes zero[35].

In order to estimate the magnitude of each remaining term and in-plane stress, we calculated $E_1$, $\partial E_1/\partial x_1$, and $\partial E_3/\partial x_3$ with electromagnetic simulation. For the simulation, a 2D model representing the cross section of the actuator in the 13-direction was developed as illustrated in the upper schematic diagram of Fig. S5. Each yellow rectangle labelled as 'top electrode' represents each tooth of the comb-shaped electrode. The resultant $E_1$, $E_3$, $\partial E_1/\partial x_1$, and $\partial E_3/\partial x_3$ values at the mid-plane and different locations on MoS$_2$ film are shown in Figs. S5 and S6. Clearly, all the electric fields and field gradients are highly concentrated at the near-edge area of the top electrode where the asymmetry occurs. $E_1$ shows the same magnitude but opposite sign at the left and right edges of the top electrode. Therefore, the tensile and compressive stresses would balance each other out, and the piezo-induced stress term $\varepsilon_{11}E_1$ becomes zero overall. Therefore, the Eq. (2) is further reduced to the Eq. (3), concluding that the actuation is solely originated from the converse flexoelectric effect.

$$\sigma_{11} = \mu_{1111}\partial E_1/\partial x_1 + \mu_{1133}\partial E_3/\partial x_3 \qquad (3)$$

To verify whether converse flexoelectricity can fully induce a 45 nm dynamic displacement, we further refined our model, as shown in Fig. 3a, to estimate the actuator's harmonic response. To reduce computational complexity and cost, the model was developed in 2D to represent the actuator's cross-section in the length-height plane, as its configuration is uniform along the width. The model includes a 50 nm-thick bottom electrode, a 500 nm-thick Parylene-C layer, a 0.65 nm

MoS$_2$ monolayer, and a comb-shaped 50 nm-thick top electrode. As fully simulating the 1-mm-long actuator with a sub-0.65 nm mesh is not feasible because of its extremely high aspect ratio and software limitations, we obtained resonant frequency and displacement trends from shorter beam lengths up to 0.20 mm and extrapolated them to estimate the behavior of the 1 mm-long actuator.

Figure 3b shows the resonant frequency trends as the actuator length changes. The resonant frequency decreases quadratically as the length increases, consistent with the theoretical relationship for beam resonance (i.e., resonant frequency $\propto 1/(\text{length})^2$). From this extrapolation, we found that the 5th-mode resonant frequency of a 1 mm beam (20.8 kHz) matches well with the actuation frequency of our device (19 kHz).

We then estimated the 5th harmonic displacement of the beam. With the assumption that the 1111-converse flexoelectric stress ($\sigma_{11} = \mu_{1111}\partial E_1/\partial x_1$, the first term in Eq. (3)) is the dominant driving mechanism of our actuator and with an estimated $\mu_{1111}$ of 1.41 nC/m, we were able to obtain displacement of 45 nm through our electro-mechanical simulation, consistent with our experimental observations. As shown in Fig. 3c, the harmonic displacement increases quadratically with the beam length, and $\mu_{1111} = 1.41$ nC/m yields 45 nm displacement for a 1 mm-long beam. Although the reported values of $\mu_{1111}$ in literature vary significantly, $\mu_{1111}$ used in our simulation lies within the reported range (0.6821-174.1 nC/m)[36–38].

To study the effects of MoS$_2$ thickness on the actuation behavior, we simulated displacement and resonance frequencies of bi- and tri-layer MoS$_2$. The resonant frequency and displacement trends remained consistent for bi- and tri-layer MoS$_2$ actuators (Fig. S7a). In addition, increasing the MoS$_2$ thickness from monolayer to bilayer reduces the harmonic displacement of the actuator, which aligns with our experimental observation that the bilayer device exhibits approximately 18% lower displacement than the monolayer counterpart. This reduction is primarily attributed to the increased flexural stiffness resulting from the additional MoS$_2$ layer.

We also evaluated displacement contributions from other potential mechanisms, including piezoelectricity, 1133-converse flexoelectric stress ($\sigma_{11} = \mu_{1133}\partial E_3/\partial x_3$, the second term in Eq. (3)), Joule heating, and electromagnetic effects. We confirmed that the 1111-converse flexoelectric effect is the dominant contributor (Supporting Information and Fig. S8). Further details regarding the simulation parameters and material properties are provided in the Methods and Supporting Information.

As our measurements of the actuator displacement were acquired at the 5th resonant frequency, there is a significant potential to further enhance the device performance by utilizing lower-order harmonic resonance. Reducing the width and pitch of the electrode combs to maximize the number of combs in the device would further improve actuation performance. Furthermore, the resonant frequency and the width of the resonant peak of the actuator can be tuned by adjusting the length, thickness, elastic modulus, and viscosity of the supporting layer to meet specific application requirements, offering a wide range of applicability.

## Stability and robustness of the actuator under extreme environment

With the understanding of the actuation mechanism and converse flexoelectric coefficients of 2D MoS$_2$, we further explored the testing of flexo-actuators under cryogenic and vacuum environment. Our actuator was operated in its resonant frequency with 40 V$_{pp}$ AC while decreasing temperatures from room temperature (RT) to 10 K in vacuum environment. We observed stable actuation performance where the actuator was able to maintain 70% of its RT displacement at 10 K (Fig. 4a). Furthermore, similar to RT behavior, our flexo-actuator exhibited linear displacement versus E-field relationship (Fig. 4b). Most interestingly, our flexo-actuator significantly outperformed a state-of-

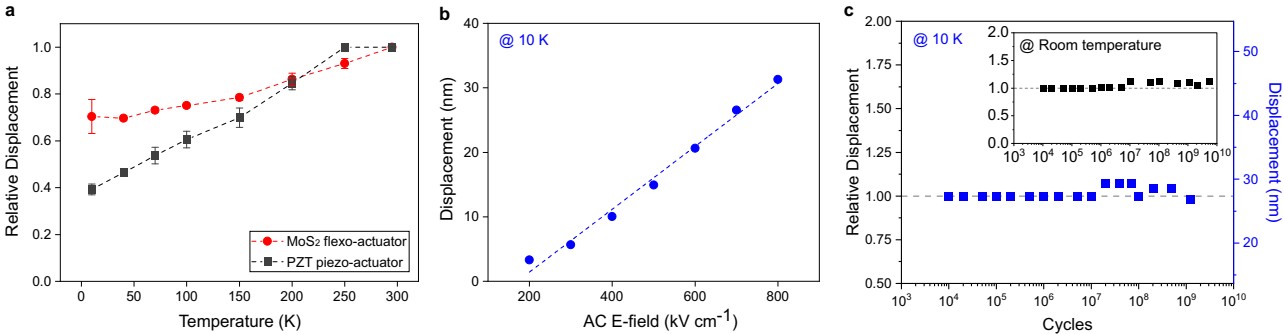

**Fig. 3 | Modeling and simulation of the converse flexoelectric MoS₂ actuator.**
**a** The 2D simulation model of a 200 μm-long converse flexo-actuator. The inset shows the monolayer MoS₂ located between the top electrode and the Parylene-C layer. For clarity, the x:y scale has been adjusted to 1:100. **b** Resonant frequency trends for actuators of different beam lengths. The 4, 5, and 6th resonant frequencies all decrease quadratically as beam length increases. Extrapolating this trend yields a 5th-mode resonance frequency of 20.8 kHz for a 1 mm actuator. **c** Fifth-harmonic displacement for various actuator lengths under 1111-converse flexoelectric stress ($\sigma_{11} = \mu_{1111}\partial E_1/\partial x_1$), using $\mu_{1111} = 1.41$ nC/m. Following this trend, a 1 mm actuator is predicted to yield a displacement of 45 nm.

**Fig. 4 | Stability and robustness of the converse flexoelectric MoS₂ actuator under extreme environment. a** Displacement of monolayer MoS₂ flexo-actuator and commercial lead zirconate titanate (PZT) piezo actuator at various temperatures down to 10 K. Error bars indicate the standard deviation of measurement data from different devices. **b** Device displacement at resonant frequency under different AC E-field at 10 K. The AC E-field was controlled by varying peak-to-peak voltage. **c** Peak displacement variation of monolayer MoS₂ flexo-actuator under continuous actuation up to $10^{10}$ cycles at both 10 K and RT.

the-art disk-type lead zirconate titanate (PZT) piezo-actuator which experienced a 60% reduction in displacement at 10 K (Fig. 4a). This result suggests flexo-actuators might serve as a robust platform for cryogenic applications where temperature scaling of piezoelectricity might pose challenges to conventional piezo-actuators[7,8,39].

To further study long-term robustness of our flexo-actuator, our monolayer $MoS_2$ device operated for $10^{10}$ cycles at both 10 K and RT. As shown in Fig. 4c, the device survived at least $10^{10}$ actuation cycles with less than 12% of performance fluctuation, regardless of operating temperature. These results support the reliability of our flexo-actuator towards cryogenic and long-term operations.

## Summary and Outlook
In conclusion, we have developed a 2D $MoS_2$ flexo-actuator based on converse flexoelectric effects of the $MoS_2$ active layer. Experimental and simulation studies demonstrated that the primary source of actuation was converse flexoelectric effect. With increasing flexo-electric effect at the nanoscale, our device exhibited performance enhanced by orders of magnitude compared to existing flexo-actuators. Additionally, the actuation of our device was linearly proportional to the applied AC voltage, allowing electrical control of actuation. Considering the extremely thin active layer thickness and performance under cryogenic conditions of our device, this study demonstrates the potential of 2D flexo-actuators for a wide range of applications which have been dominated by piezo-actuators.

## Methods
### Device fabrication
To fabricate our device, we prepared a multilayer beam structure of $Au/MoS_2/Parylene-C/Ag$ on a 300 nm $SiO_2/Si$ wafer and then transferred it onto a sample holder. A 5 wt% polyacrylic acid (PAA) solution was spin-coated to form a thin sacrificial layer on the $SiO_2$ wafer. A 50 nm thick bottom Ag electrode was then deposited by electron beam (E-beam) evaporator. For mechanical support and electrical insulation, a 500 nm thick Parylene-C layer was coated by Parylene deposition system (PDS 2010). Then, a monolayer $MoS_2$ film grown on a $SiO_2/Si$ wafer by metal-organic chemical vapor deposition (MOCVD) was transferred onto the Parylene-C layer. We placed the $MoS_2/SiO_2/Si$ wafer upside down onto the Parylene-C layer and dropped a water droplet to separate monolayer $MoS_2$ from the $SiO_2/Si$ wafer by water penetration. After we fully dried the $MoS_2/Parylene-C/Ag/PAA/SiO_2/Si$ sample at ambient condition, we coated a patterned photoresist layer onto the sample through photolithography. Then, a 50 nm thick Au top electrode was deposited by E-beam evaporator. The photoresist layer was removed after the deposition. To transfer the $Au/MoS_2/Parylene-C/Ag$ layer from the wafer to a sample holder, the PAA layer was dissolved by placing the sample on water surface. After full separation, the four-layer structure was transferred onto the sample holder. The sample holder was prepared by placing two glass slides having a 1 mm gap between them.

### Simulation setup
COMSOL Multiphysics was employed to investigate the actuation mechanism of our device. The model was developed in a 2D plane and consisted of four layers: a 50 nm-thick bottom electrode, a 500 nm-thick Parylene-C layer, a $MoS_2$ layer with thicknesses of 0.65, 1.3, or 1.95 nm depending on the number of layers, and a 50 nm-thick top electrode. The top electrode was designed with 25 individual rectangular segments to mimic a comb-shaped electrode in 2D.

Electric field and electric field gradient distributions within the $MoS_2$ layer were computed using the Electrostatics module. Effects of flexoelectricity and piezoelectricity on the device actuation were conducted using the Electrostatics, Solid Mechanics, and Piezoelectric Effect Multiphysics modules in combination. Since the flexoelectric effect is not natively supported in COMSOL, the equations for the

variables solid.PpzeX (Piezoelectric polarization, X component) and solid.Spze11 (Piezoelectric stress tensor, local coordinate system, 11 component) were modified by incorporating a $\mu_{1111}$- and $\mu_{1133}$-dependent terms into each expression, mu_1111*d(solid.eel11,x) + mu_1133*d(solid.eel22,y) and -mu_1111*d(solid.EpzeX,x) - mu_1133*d(-solid.EpzeY,y), respectively. Note that y represents 3-directional coordinate as our model was built in 2D. Furthermore, electromagnetic stress calculations were performed using the Electrostatics, Solid Mechanics, and Electromechanical Forces Multiphysics modules based on the Minkowski tensor to estimate the device deformation due to electromagnetic forces under the applied AC voltage. Joule heating effects were also simulated to calculate the device actuation by thermal expansion resulting from heat generated in the top and bottom electrodes, using the Electric Currents, Heat Transfer in Solids, Electromagnetic Heating, and Thermal Expansion Multiphysics modules.

## Data availability
The data that support the plots within this paper and other findings of this study are available at https://doi.org/10.5061/dryad.xksn02vwc.

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

## Acknowledgements

We gratefully acknowledge support from the NASA (NNX16AR56G and 80NSSC25K7278) (S.N.), ONR (N00014-17-1-2830 and N000142412533) (S.N.), NSF (CMMI-2135734, CMMI-2306039, and CBET-2035584) (S.N.), and National Research Foundation of Korea (NRF) funded by Ministry of Science and ICT (RS-2024-00408180) (S.N.). Experiments were carried out in part in the Materials Research Laboratory Central Research Facilities, Beckman Institute, and Holonyak Micro and Nano Technology Laboratory at the University of Illinois at Urbana-Champaign.

## Author contributions

Y.L. and H.J.B. contributed equally to this work. H.J.B. designed the device and Y.L., H.J.B., K.-H.L., and J.M.K. contributed to device fabrication. Y.L., H.J.B., and M.F.H. performed device characterizations, and Y.L. and W.G. conducted the simulations. S.N. conceived and supervised the work. The manuscript was written by Y.L., H.J.B., W.G., and S.N. with comments and inputs from all authors.

## Competing interests

The authors declare no competing interests.
