## [Transparent Peer Review file · Nature Communications]

Converse Flexoelectric Two-dimensional MoS₂ Actuator

Corresponding Author: Professor SungWoo Nam

Version 0:

Reviewer comments:

Reviewer #1

(Remarks to the Author)

The authors present a MoS₂ based flexoelectric actuation device, with significantly high response and durability under different thermal and mechanical conditions. The results are important, and the reported device holds potential in nano-actuation and control. The manuscript can be considered for publication after addressing the following major issues:

1. The authors claim to report the first 3D material-based flexoelectric actuator. However, the work by Hirakata et al (DOI: 10.1088/1361-6463/ac4367) also reports a similar actuator made of MoS₂. Authors should clarify the difference between their work and Hirakata et al. and justify the novelty argument.
2. The measurements and computations presented are mostly derived from indirect calculations. For example, it appears that the calculations of stress based on the displacement and 2D electromagnetic simulations are performed assuming only elasticity, not the coupled electromechanical physics. If a 2-way electromechanical coupling is indeed considered, the boundary conditions and assumptions should be reported accurately. It is crucial to consider the effect of fully coupled electromechanical effects since it can have a significant impact on the final results at such small scales.
3. Line 69-71: "This actuation level is more than an order of magnitude higher than that of reported flexoelectric actuators when normalized by active layer thickness and applied voltage, and it is even comparable to piezo-actuators." The source of comparison should be cited if the authors are comparing their results with existing literature.
4. Figure S2 just shows a screenshot of the oscilloscope measurements without axis labels or any form of quantification. This does not contribute anything to the results and raises concern about the measurements. Please replace it with proper plots of input voltage signals, output laser vibrometer measurements, and the FFT transforms, which would be very useful for reproducibility of the results.
5. The reported gain in actuation with MoS₂ vs without MoS₂ layer (control) is not conclusive enough to credit the origin of actuation to flexoelectricity. The other plausible reasons for this could be a) Maxwell stress could enhance the actuation due to local fringe effects introduced by the MoS₂ layer. b) thermoelastic bending can occur due to Joule heating caused by heat dissipation in the metal combs. These effects can be eliminated by introducing an inert layer or a layer with a known flexoelectric effect. Moreover, the mechanical resonance gains should also be eliminated by comparing the control and MoS₂ layered samples off-resonance.

Reviewer #2

(Remarks to the Author)

The work is just fine and should be published "somewhere" but I see no substantive novelty that would warrant the readership of Nature Communications. Flexoelectricity has been well-studied theoretically and experimentally. Including in 2D materials. It would be like for me to publish a paper on a piezoelectric crystal and claim that it acts like an actuator! Of course it can and will. I don't see what the big deal of the current paper is other than busy work. It would, for example, be a perfectly fine paper for any number of lower tier journals.

Reviewer #3

(Remarks to the Author)

This manuscript presents a highly novel and compelling study on the development of a converse flexoelectric actuator utilizing monolayer MoS₂ as the active material. The combination of experimental characterization, electromagnetic simulation, and mechanical modeling provides a multi-faceted and convincing argument for the converse flexoelectric

actuation mechanism. I recommend acceptance after addressing the minor points outlined below.

1. The estimated range for μ_{1111} and μ_{1133} (0.1–0.2 nC/m) is notably lower than some values reported in the literature. While the authors correctly state that reported values vary widely, a brief discussion on the potential reasons for this discrepancy would strengthen the manuscript.

2. The actuation performance of monolayer and bilayer devices is similar. The deep reason should be reached. Is it purely flexoelectricity? If so, does the interface between layers or the slip plane play a role?

3. The actuator demonstrates impressive performance at its resonant frequency (~19 kHz). However, for many practical applications, a broader frequency band is highly desirable. Could the authors comment on the expected performance of the device when operated off-resonance? What potential strategies could be explored to achieve a wider bandwidth?

Version 1:

Reviewer comments:

Reviewer #1

(Remarks to the Author)

The authors have performed a rigorous revision of the manuscript, addressing all of the comments carefully. Therefore, I recommend the manuscript for publication in its current form.

Reviewer #3

(Remarks to the Author)

The authors have addressed all my concerns. I recommend this work for publication now.

Reviewer #1 (Remarks to the Author):

The authors present a MoS₂ based flexoelectric actuation device, with significantly high response and durability under different thermal and mechanical conditions. The results are important, and the reported device holds potential in nano-actuation and control. The manuscript can be considered for publication after addressing the following major issues:

We thank the reviewer for their thoughtful and constructive feedback, as well as for recognizing the significance and potential of our work. We greatly appreciate the positive evaluation of the device performance and its relevance to nano-actuation and control applications. We have carefully addressed each of the major issues raised and provided a detailed, point-by-point response to the comments below.

1. The authors claim to report the first 3D material-based flexoelectric actuator. However, the work by Hirakata et al (DOI: 10.1088/1361-6463/ac4367) also reports a similar actuator made of MoS₂. Authors should clarify the difference between their work and Hirakata et al. and justify the novelty argument.

We thank the referee for this comment. The work by Hirakata et al. reported out-of-plane deformation of MoS₂ at the materials level, not the device scale. It turns out that most recent papers on the flexoelectricity of 2D materials are either computational, limited to the materials level, or focus on out-of-plane directional strain. In contrast, our actuator uses in-plane stress to actuate the entire device. To the best of our knowledge, this is the first experimentally demonstrated 2D material-based flexoelectric actuator. We have included the paper by Hirakata et al. to the references and included additional discussion in the main text to clarify our novelty, as shown below.

Revision to MS/SI

In main text, page 4:

~. Although the flexoelectric effect of 2D materials has been ~~studied for its fundamental properties~~ explored in both theoretical and experimental studies, most works have focused on materials-level behavior or computational modeling^{12,17-19}. In contrast, an actuator driven by converse-flexoelectric response of 2D materials has not been demonstrated to the best of our knowledge.

Here we show a flexo-actuator operated by converse-flexoelectric effect leveraging 2D materials. ~(text omitted)~ Furthermore, our device shows robust performance up to 10¹⁰ cycles and maintains ~70% of its actuation under 10 K cryogenic condition in contrast to the conventional piezoelectric actuator maintaining only ~40% of its actuation under the same condition. Our results highlight the novelty of flexoelectricity in 2D materials, demonstrating substantial scaling advantages and strong potential for cryogenic device applications.

In References:

19. Hirakata, H., Fukuda, Y. & Shimada, T. Flexoelectric properties of multilayer two-dimensional material MoS₂. *J Phys D Appl Phys* **55**, 125302 (2021).

2. The measurements and computations presented are mostly derived from indirect calculations. For example, it appears that the calculations of stress based on the displacement and 2D electromagnetic simulations are performed assuming only elasticity, not the coupled electromechanical physics. If a 2-way electromechanical coupling is indeed considered, the boundary conditions and assumptions should be reported accurately. It is crucial to consider the effect of fully coupled electromechanical effects since it can have a significant impact on the final results at such small scales.

We thank the referee for the valuable insights. We have modified the electromechanical model to include flexoelectricity and conducted full simulations. Also, we have included these new results in our revised manuscript as well as in the Methods and Supporting Information as shown below.

Revision to MS/SI

In main text, page 8:

To verify whether converse flexoelectricity can fully induce a 45 nm dynamic displacement, we further refined our model, as shown in Figure 3a, to estimate the actuator's harmonic response. To reduce computational complexity and cost, the model was developed in 2D to represent the actuator's cross-section in the length-height plane, as its configuration is uniform along the width. The model includes a 50 nm-thick bottom electrode, a 500 nm-thick Parylene-C layer, a 0.65 nm MoS₂ monolayer, and a comb-shaped 50 nm-thick top electrode. As fully simulating the 1-mm-long actuator with a sub-0.65-nm mesh is not feasible because of its extremely high aspect ratio and software limitations, we obtained resonant frequency and displacement trends from shorter beam lengths up to 0.20 mm and extrapolated them to estimate the behavior of the 1 mm-long actuator.

Figure 3b shows the resonant frequency trends as the actuator length changes. The resonant frequency decreases quadratically as the length increases, consistent with the theoretical relationship for beam resonance (i.e., resonant frequency $\propto 1/(\text{length})^2$). From this extrapolation, we found that the 5th-mode resonant frequency of a 1 mm beam (20.8 kHz) matches well with the actuation frequency of our device (19 kHz).

We then estimated the 5th harmonic displacement of the beam. With the assumption that the 1111-converse flexoelectric stress ($\sigma_{11} = \mu_{1111}\partial E_1/\partial x_1$, the first term in equation (3)) is the dominant driving mechanism of our actuator and with an estimated μ_{1111} of 1.41 nC/m, we were able to obtain displacement of 45 nm through our electromechanical simulation, consistent with our experimental observations. As shown in Figure 3c, the harmonic displacement increases quadratically with the beam length, and $\mu_{1111} = 1.41$ nC/m yields 45 nm displacement for a 1 mm-long beam. Although the reported values of μ_{1111} in literature vary significantly, μ_{1111} used in our simulation lies within the reported range (0.6821-174.1 nC/m³⁷⁻³⁹).

To study the effects of MoS₂ thickness on the actuation behavior, we simulated displacement and resonance frequencies of bi- and tri-layer MoS₂. The resonant frequency and displacement trends remained consistent for bi- and tri-layer MoS₂ actuators (Figure S7a). In addition, increasing the MoS₂ thickness from monolayer to bilayer reduces the harmonic displacement of the actuator, which aligns with our experimental observation that the bilayer device exhibits approximately 18% lower displacement than the monolayer counterpart. This reduction is primarily attributed to the increased flexural stiffness resulting from the additional MoS₂ layer.

We also evaluated displacement contributions from other potential mechanisms, including piezoelectricity, 1133-converse flexoelectric stress ($\sigma_{11} = \mu_{1133}\partial E_3/\partial x_3$, the second term in equation (3)), Joule heating, and electromagnetic effects. We confirmed that the 1111-converse flexoelectric effect is the dominant contributor (Supporting Information and Figure S8). Further details regarding the simulation parameters and material properties are provided in the Methods and Supporting Information.

Figure 3:

Figure 3. (a) The 2D simulation model of a 200 μm-long converse-flexoelectric actuator. The upper-left inset shows the monolayer MoS₂ located between the top electrode and the Parylene-C layer. For clarity, the x:y scale has been adjusted to 1:100. (b) Resonant frequency trends for actuators of different beam lengths. The 4th, 5th, and 6th resonant frequencies all decrease quadratically as beam length increases. Extrapolating this trend yields a 5th-mode resonance frequency of 20.8 kHz for a 1 mm actuator. (c) Fifth-harmonic displacement for various actuator lengths under 1111-converse-flexoelectric stress ($\sigma_{11} = \mu_{1111}\partial E_1/\partial x_1$), using $\mu_{1111} = 1.41$ nC/m. Following this trend, a 1 mm actuator is predicted to yield a displacement of 45 nm.

In Methods:

(2) Simulation set-up

COMSOL Multiphysics was employed to investigate the actuation mechanism of our device. The model was developed in a 2D plane and consisted of four layers: a 50 nm-thick bottom electrode,

a 500 nm-thick Parylene-C layer, a MoS₂ layer with thicknesses of 0.65, 1.3, or 1.95 nm depending on the number of layers, and a 50 nm-thick top electrode. The top electrode was designed with 25 individual rectangular segments to mimic a comb-shaped electrode in 2D.

Electric field and electric field gradient distributions within the MoS₂ layer were computed using the Electrostatics module. Effects of flexoelectricity and piezoelectricity on the device actuation were conducted using the Electrostatics, Solid Mechanics, and Piezoelectric Effect Multiphysics modules in combination. Since the flexoelectric effect is not natively supported in COMSOL, the equations for the variables solid.PpzeX (Piezoelectric polarization, X component) and solid.Spze11 (Piezoelectric stress tensor, local coordinate system, 11 component) were modified by incorporating a μ_{1111} - and μ_{1133} -dependent terms into each expression, $\mu_{1111} * d(\text{solid.eel11}, x) + \mu_{1133} * d(\text{solid.eel22}, y)$ and $-\mu_{1111} * d(\text{solid.EpzeX}, x) - \mu_{1133} * d(\text{solid.EpzeY}, y)$, respectively. Note that y represents 3-directional coordinate as our model was built in 2D. Furthermore, electromagnetic stress calculations were performed using the Electrostatics, Solid Mechanics, and Electromechanical Forces Multiphysics modules based on the Minkowski tensor to estimate the device deformation due to electromagnetic forces under the applied AC voltage. Joule heating effects were also simulated to calculate the device actuation by thermal expansion resulting from heat generated in the top and bottom electrodes, using the Electric Currents, Heat Transfer in Solids, Electromagnetic Heating, and Thermal Expansion Multiphysics modules.

In Supporting Information:

(1) Calculation of Q factor for including damping in the simulation

In order to accurately calculate the harmonic displacement, we first estimated the Q factor from experimental data and apply it to the simulation. Figure S9 shows the frequency-displacement plot near the 5th resonant frequency. The Q factor was calculated by dividing resonant frequency by the width of the peak at its half height (Δf), which is $20,700 \text{ Hz} / 690 \text{ Hz} = 30$.

(2) Estimation of harmonic displacement of mono-, bi-, and tri-layer MoS₂ actuators

While we were able to simulate up to 0.20 mm for monolayer MoS₂ actuators due to aspect ratio limitations, the thicker MoS₂ layer enables us to simulate the resonant frequencies and harmonic displacement of the longer beam. Figure S7a and b show the 5th harmonic frequency and displacement of mono-, bi-, and tri-layer MoS₂ actuators with different lengths. As with the monolayer case, bi- and tri-layer devices also show quadratically decreasing resonant frequency and quadratically increasing harmonic displacement as the beam length increases. In addition, bilayer MoS₂ actuator shows lower harmonic displacement compared to the monolayer MoS₂, agreeing well with our experimental results.

(3) Estimation of actuation from piezoelectric effect, 1133-converse flexoelectric stress, electromagnetic stress, and Joule heating

Using the 200 μm -long model with monolayer MoS₂, we also calculated the contributions of other possible mechanisms to the 5th harmonic displacement, including the piezoelectric effect, 1133-converse flexoelectric stress, electromagnetic stress, and Joule heating. In case of the 1133-converse flexoelectric stress, we applied 1.41 nC/m for μ_{1133} , the same value as the estimated μ_{1111} value. Table S1 summarizes the parameters used in each simulation, and Figure S8 illustrates the device actuation resulting from each effect. The maximum displacements obtained from the 1111- and 1133-converse flexoelectric effects, piezoelectric effect, and electromagnetic stress are 1.64, 0.014, 0.30, and 0.29 nm, respectively. The displacement resulting from Joule heating is more than 10 orders of magnitude smaller than those from the other effects and can therefore be neglected. These results strongly support that the 1111-converse flexoelectric effect is the dominant mechanism driving device actuation, unless monolayer MoS₂ exhibits a μ_{1133} value that is significantly higher than μ_{1111} . Additionally, when these other effects are taken into consideration, μ_{1111} is estimated to be in the range of 1.41 ± 0.52 nC/m, under the assumption that $\mu_{1111} = \mu_{1133}$.

Table S1. Simulation parameters

Module	Parameter	Value
Solid Mechanics	Young's modulus, Au	70 GPa
	Poisson's ratio, Au	0.44
	Young's modulus, Ag	83 GPa
	Poisson's ratio, Ag	0.37
	Young's modulus, Parylene-C	2.75 GPa
	Poisson's ratio, Parylene-C	0.4
	Elastic modulus, C ₁₁ , MoS ₂	128.4 GPa ^[1]
	Elastic modulus, C ₂₂ , MoS ₂	128.4 GPa ^[1]
	Elastic modulus, C ₁₂ , MoS ₂	32.6 GPa ^[1]
	Elastic modulus, C ₃₃ , MoS ₂	38.3 GPa ^[2]
	Elastic modulus, C ₄₄ , MoS ₂	26.5 GPa ^[2]
	Elastic modulus, C ₅₅ , MoS ₂	26.5 GPa ^[2]
	Elastic modulus, C ₆₆ , MoS ₂	47.9 GPa ((C ₁₁ -C ₁₂)/2)
	Flexoelectric coefficient, μ_{1111} , MoS ₂	1.41 nC/m

	Flexoelectric coefficient, μ_{1133} , MoS ₂	1.41 nC/m
	Piezoelectric coefficient, e_{11} , MoS ₂	0.5 C/m ² [3]
	Isotropic structural loss factor (1/Q factor)	1/30
Electrostatics & Electric Currents	Terminal voltage, top electrode (Au)	20 V
	Terminal voltage, bottom electrode (Ag)	0 V
	Relative permittivity, Parylene-C	3
	Relative permittivity, MoS ₂	4 [4]
Heat Transfer in Solids	Thermal conductivity, Au	317 W/m·K
	Density, Au	19300 kg/m ³
	Heat capacity, Au	129 J/kg·K
	Thermal conductivity, Ag	429 W/m·K
	Density, Ag	10500 kg/m ³
	Heat capacity, Ag	235 J/kg·K
	Thermal conductivity, Parylene-C	0.084 W/m·K
	Density, Parylene-C	1289 kg/m ³
	Heat capacity, Parylene-C	712 J/kg·K
	Thermal conductivity, MoS ₂	23.2 W/m·K [5]
	Density, MoS ₂	5060 kg/m ³
	Heat capacity, MoS ₂	87.5 J/kg·K (14 J/mol·K) [6]
	Thermal Expansion (Multiphysics)	Thermal expansion coefficient, Au
Thermal expansion coefficient, Ag		1.89x10 ⁻⁵ /K
Thermal expansion coefficient, Parylene-C		3.5x10 ⁻⁵ /K
Thermal expansion coefficient, MoS ₂		7.6x10 ⁻⁶ /K [7]

Supplementary Figures:

Figure S7. (a) Resonant frequency trends for actuators with different MoS₂ layer thicknesses. A quadratically decreasing trend is maintained as the MoS₂ layer becomes thicker with increasing layer number. (b) Fifth-harmonic displacement for different MoS₂ layer thicknesses. Although the trend is maintained, the bi- and tri-layer MoS₂ actuators exhibit lower displacement compared to the monolayer MoS₂ actuator.

Figure S8. Fifth-harmonic displacement resulting from the 1111- and 1133-converse flexoelectric effects, piezoelectric effect, electromagnetic stress, and Joule heating. A 200 μm -long model with monolayer MoS_2 was implemented for the calculations, for which the 5th resonant frequency is estimated to be ~ 430 kHz.

Figure S9. Fifth-harmonic displacement of the monolayer MoS_2 actuator near the resonant frequency range for Q-factor calculation.

3. Line 69-71: “This actuation level is more than an order of magnitude higher than that of reported flexoelectric actuators when normalized by active layer thickness and applied voltage, and it is even comparable to piezo-actuators.” The source of comparison should be cited if the authors are comparing their results with existing literature.

We appreciate the referee’s thorough review of our manuscript. We have added citations to the sentence.

Revision to MS/SI:

This actuation level is more than an order of magnitude higher than that of reported flexoelectric actuators when normalized by active layer thickness and applied voltage^{13–15}, and it is even comparable to piezo-actuators^{20–30}.

4. Figure S2 just shows a screenshot of the oscilloscope measurements without axis labels or any form of quantification. This does not contribute anything to the results and raises concern about the measurements. Please replace it with proper plots of input voltage signals, output laser

vibrometer measurements, and the FFT transforms, which would be very useful for reproducibility of the results.

We thank the referee for this comment. We have updated Figure S2 to the following figure.

Figure S2. (a) AC voltage applied to the device at the 5th harmonic frequency for actuation. (b) Output voltage signal from the laser vibrometer after fast Fourier transform (FFT), which is converted to vertical velocity and then to displacement.

5. The reported gain in actuation with MoS2 vs without MoS2 layer (control) is not conclusive enough to credit the origin of actuation to flexoelectricity. The other plausible reasons for this could be a) Maxwell stress could enhance the actuation due to local fringe effects introduced by the MoS2 layer. b) thermoelastic bending can occur due to Joule heating caused by heat dissipation in the metal combs. These effects can be eliminated by introducing an inert layer or a layer with a known flexoelectric effect. Moreover, the mechanical resonance gains should also be eliminated by comparing the control and MoS2 layered samples off-resonance.

We appreciate the referee's valuable comment. We examined other possible effects as suggested and modified the manuscript and Supporting Information as follows:

Revision to MS/SI

In main text, page 10:

We also evaluated displacement contributions from other potential mechanisms, including piezoelectricity, 1133-converse flexoelectric stress ($\sigma_{11} = \mu_{1133}\partial E_3/\partial x_3$, the second term in equation (3)), Joule heating, and electromagnetic effects. We confirmed that the 1111-converse flexoelectric effect is the dominant contributor (Supporting Information and Figure S8). Further details regarding the simulation parameters and material properties are provided in the Methods

and Supporting Information.

In Supporting Information:

(3) Estimation of actuation from piezoelectric effect, 1133-converse flexoelectric stress, electromagnetic stress, and Joule heating

Using the 200 μm -long model with monolayer MoS_2 , we also calculated the contributions of other possible mechanisms to the 5th harmonic displacement, including the piezoelectric effect, 1133-converse flexoelectric stress, electromagnetic stress, and Joule heating. In case of the 1133-converse flexoelectric stress, we applied 1.41 nC/m for μ_{1133} , the same value as the estimated μ_{1111} value. Table S1 summarizes the parameters used in each simulation, and Figure S8 illustrates the device actuation resulting from each effect. The maximum displacements obtained from the 1111- and 1133-converse flexoelectric effects, piezoelectric effect, and electromagnetic stress are 1.64, 0.014, 0.30, and 0.29 nm, respectively. The displacement resulting from Joule heating is more than 10 orders of magnitude smaller than those from the other effects and can therefore be neglected. These results strongly support that the 1111-converse flexoelectric effect is the dominant mechanism driving device actuation, unless monolayer MoS_2 exhibits a μ_{1133} value that is significantly higher than μ_{1111} . Additionally, when these other effects are taken into consideration, μ_{1111} is estimated to be in the range of 1.41 ± 0.52 nC/m, under the assumption that $\mu_{1111} = \mu_{1133}$.

Figure S8:

Figure S8. Fifth-harmonic displacement resulting from the 1111- and 1133-converse flexoelectric effect, piezoelectric effect, electromagnetic stress, and Joule heating. A 200 μm -long model with monolayer MoS_2 was used for the calculations, for which the 5th resonant frequency is estimated to be ~ 430 kHz.

Reviewer #2 (Remarks to the Author):

The work is just fine and should be published "somewhere" but I see no substantive novelty that would warrant the readership of Nature Communications. Flexoelectricity has been well-studied theoretically and experimentally. Including in 2D materials. It would be like for me to publish a paper on a piezoelectric crystal and claim that it acts like an actuator! Of course it can and will. I don't see what the big deal of the current paper is other than busy work. It would, for example, be a perfectly fine paper for any number of lower tier journals.

We appreciate the referee's time and thoughtful critique. We would like to take this opportunity to further clarify the specific novelty and contribution of our work in the context of existing literature.

While flexoelectricity in 2D materials has been explored in both theoretical and experimental studies, most of prior works have focused on materials-level phenomena, computational modeling, or out-of-plane deformation modes. In contrast, our study presents what we believe to be the first experimental demonstration of flexoelectricity using a 2D material at the device level for out-of-plane actuation. Through this demonstration, we reveal the substantial scaling advantages of flexoelectric effect in 2D materials and their strong potential for cryogenic device applications. This is a distinct departure from previous work and, in our view, represents a meaningful step toward translating fundamental material properties into functional actuators. By bridging the gap between theoretical predictions and practical implementation, we aim to provide a proof-of-concept that 2D flexoelectric materials can serve as the foundation for next-generation actuation systems - a message we believe aligns with the interdisciplinary and application-oriented scope of *Nature Communications*.

We hope these points help clarify the contribution of our work and its relevance to the journal's readership, and we added the following sentences in the manuscript:

Revision to MS/SI

In main text, page 4:

~. Although the flexoelectric effect of 2D materials has been ~~studied for its fundamental properties~~ explored in both theoretical and experimental studies, most works have focused on materials-level behavior or computational modeling^{12,17-19}. In contrast, an actuator driven by converse-flexoelectric response of 2D materials has not been demonstrated to the best of our knowledge.

Here we show a flexo-actuator operated by converse-flexoelectric effect leveraging 2D materials. ~(text omitted)~ Furthermore, our device shows robust performance up to 10^{10} cycles and maintains ~70% of its actuation under 10 K cryogenic condition in contrast to the conventional piezoelectric actuator maintaining only ~40% of its actuation under the same condition. Our results highlight the novelty of flexoelectricity in 2D materials, demonstrating substantial scaling advantages and strong potential for cryogenic device applications.

Reviewer #3 (Remarks to the Author):

This manuscript presents a highly novel and compelling study on the development of a converse flexoelectric actuator utilizing monolayer MoS₂ as the active material. The combination of experimental characterization, electromagnetic simulation, and mechanical modeling provides a multi-faceted and convincing argument for the converse flexoelectric actuation mechanism. I recommend acceptance after addressing the minor points outlined below.

We sincerely thank the reviewer for their positive and encouraging feedback on our work. We are glad that the novelty and multi-faceted approach of our study were appreciated. We have carefully addressed all the minor points raised and have made the corresponding revisions in the manuscript, as detailed in our point-by-point response below.

1. The estimated range for μ_{1111} and μ_{1133} (0.1–0.2 nC/m) is notably lower than some values reported in the literature. While the authors correctly state that reported values vary widely, a brief discussion on the potential reasons for this discrepancy would strengthen the manuscript.

We thank the referee for the valuable advice. We have thoroughly revised our model with a fully coupled electromechanical model for more accurate calculations and found that the measured deformation arises from a higher-order harmonic rather than from the first harmonic excitation. The resulting estimated flexoelectric coefficient (1.41 nC/m) has been revised accordingly and now falls within the theoretically reported range (0.6821–174.1 nC/m). Potential discrepancies between our estimation and the actual value would mainly stem from the actual elastic properties and permittivity of MoS₂ and Parylene-C in the high-frequency range. We have revised our manuscript and Supporting Information as follows:

Revision to MS/SI

In main text, page 8:

To verify whether converse flexoelectricity can fully induce a 45 nm dynamic displacement, we further refined our model, as shown in Figure 3a, to estimate the actuator's harmonic response. To reduce computational complexity and cost, the model was developed in 2D to represent the actuator's cross-section in the length-height plane, as its configuration is uniform along the width. The model includes a 50 nm-thick bottom electrode, a 500 nm-thick Parylene-C layer, a 0.65 nm MoS₂ monolayer, and a comb-shaped 50 nm-thick top electrode. As fully simulating the 1-mm-long actuator with a sub-0.65-nm mesh is not feasible because of its extremely high aspect ratio and software limitations, we obtained resonant frequency and displacement trends from shorter beam lengths up to 0.20 mm and extrapolated them to estimate the behavior of the 1 mm-long actuator.

Figure 3b shows the resonant frequency trends as the actuator length changes. The resonant frequency decreases quadratically as the length increases, consistent with the theoretical relationship for beam resonance (i.e., resonant frequency $\propto 1/(\text{length})^2$). From this extrapolation, we found that the 5th-mode resonant frequency of a 1 mm beam (20.8 kHz) matches well with

the actuation frequency of our device (19 kHz).

We then estimated the 5th harmonic displacement of the beam. With the assumption that the 1111-converse flexoelectric stress ($\sigma_{11} = \mu_{1111}\partial E_1/\partial x_1$, the first term in equation (3)) is the dominant driving mechanism of our actuator and with an estimated μ_{1111} of 1.41 nC/m, we were able to obtain displacement of 45 nm through our electromechanical simulation, consistent with our experimental observations. As shown in Figure 3c, the harmonic displacement increases quadratically with the beam length, and $\mu_{1111} = 1.41$ nC/m yields 45 nm displacement for a 1 mm-long beam. Although the reported values of μ_{1111} in literature vary significantly, μ_{1111} used in our simulation lies within the reported range (0.6821-174.1 nC/m³⁷⁻³⁹).

To study the effects of MoS₂ thickness on the actuation behavior, we simulated displacement and resonance frequencies of bi- and tri-layer MoS₂. The resonant frequency and displacement trends remained consistent for bi- and tri-layer MoS₂ actuators (Figure S7a). In addition, increasing the MoS₂ thickness from monolayer to bilayer reduces the harmonic displacement of the actuator, which aligns with our experimental observation that the bilayer device exhibits approximately 18% lower displacement than the monolayer counterpart. This reduction is primarily attributed to the increased flexural stiffness resulting from the additional MoS₂ layer.

We also evaluated displacement contributions from other potential mechanisms, including piezoelectricity, 1133-converse flexoelectric stress ($\sigma_{11} = \mu_{1133}\partial E_3/\partial x_3$, the second term in equation (3)), Joule heating, and electromagnetic effects. We confirmed that the 1111-converse flexoelectric effect is the dominant contributor (Supporting Information and Figure S8). Further details regarding the simulation parameters and material properties are provided in the Methods and Supporting Information.

Figure 3:

Figure 3. (a) The 2D simulation model of a 200 μm -long converse-flexoelectric actuator. The upper-left inset shows the monolayer MoS_2 located between the top electrode and the Parylene-C layer. For clarity, the x:y scale has been adjusted to 1:100. (b) Resonant frequency trends for actuators of different beam lengths. The 4th, 5th, and 6th resonant frequencies all decrease quadratically as beam length increases. Extrapolating this trend yields a 5th-mode resonance frequency of 20.8 kHz for a 1 mm actuator. (c) Fifth-harmonic displacement for various actuator lengths under 1111-converse-flexoelectric stress ($\sigma_{11} = \mu_{1111}\partial E_1/\partial x_1$), using $\mu_{1111} = 1.41 \text{ nC/m}$. Following this trend, a 1 mm actuator is predicted to yield a displacement of 45 nm.

In Methods:

(2) Simulation set-up

COMSOL Multiphysics was employed to investigate the actuation mechanism of our device. The model was developed in a 2D plane and consisted of four layers: a 50 nm-thick bottom electrode,

a 500 nm-thick Parylene-C layer, a MoS₂ layer with thicknesses of 0.65, 1.3, or 1.95 nm depending on the number of layers, and a 50 nm-thick top electrode. The top electrode was designed with 25 individual rectangular segments to mimic a comb-shaped electrode in 2D.

Electric field and electric field gradient distributions within the MoS₂ layer were computed using the Electrostatics module. Effects of flexoelectricity and piezoelectricity on the device actuation were conducted using the Electrostatics, Solid Mechanics, and Piezoelectric Effect Multiphysics modules in combination. Since the flexoelectric effect is not natively supported in COMSOL, the equations for the variables solid.PpzeX (Piezoelectric polarization, X component) and solid.Spze11 (Piezoelectric stress tensor, local coordinate system, 11 component) were modified by incorporating a μ_{1111} - and μ_{1133} -dependent terms into each expression, $\mu_{1111} * d(\text{solid.eel11}, x) + \mu_{1133} * d(\text{solid.eel22}, y)$ and $-\mu_{1111} * d(\text{solid.EpzeX}, x) - \mu_{1133} * d(\text{solid.EpzeY}, y)$, respectively. Note that y represents 3-directional coordinate as our model was built in 2D. Furthermore, electromagnetic stress calculations were performed using the Electrostatics, Solid Mechanics, and Electromechanical Forces Multiphysics modules based on the Minkowski tensor to estimate the device deformation due to electromagnetic forces under the applied AC voltage. Joule heating effects were also simulated to calculate the device actuation by thermal expansion resulting from heat generated in the top and bottom electrodes, using the Electric Currents, Heat Transfer in Solids, Electromagnetic Heating, and Thermal Expansion Multiphysics modules.

In Supporting Information:

(1) Calculation of Q factor for including damping in the simulation

In order to accurately calculate the harmonic displacement, we first estimated the Q factor from experimental data and apply it to the simulation. Figure S9 shows the frequency-displacement plot near the 5th resonant frequency. The Q factor was calculated by dividing resonant frequency by the width of the peak at its half height (Δf), which is 20,700 Hz / 690 Hz = 30.

(2) Estimation of harmonic displacement of mono-, bi-, and tri-layer MoS₂ actuators

While we were able to simulate up to 0.20 mm for monolayer MoS₂ actuators due to aspect ratio limitations, the thicker MoS₂ layer enables us to simulate the resonant frequencies and harmonic displacement of the longer beam. Figure S7a and b show the 5th harmonic frequency and displacement of mono-, bi-, and tri-layer MoS₂ actuators with different lengths. As with the monolayer case, bi- and tri-layer devices also show quadratically decreasing resonant frequency and quadratically increasing harmonic displacement as the beam length increases. In addition, bilayer MoS₂ actuator shows lower harmonic displacement compared to the monolayer MoS₂, agreeing well with our experimental results.

(3) Estimation of actuation from piezoelectric effect, 1133-converse flexoelectric stress, electromagnetic stress, and Joule heating

Using the 200 μm -long model with monolayer MoS₂, we also calculated the contributions of other possible mechanisms to the 5th harmonic displacement, including the piezoelectric effect, 1133-converse flexoelectric stress, electromagnetic stress, and Joule heating. In case of the 1133-converse flexoelectric stress, we applied 1.41 nC/m for μ_{1133} , the same value as the estimated μ_{1111} value. Table S1 summarizes the parameters used in each simulation, and Figure S8 illustrates the device actuation resulting from each effect. The maximum displacements obtained from the 1111- and 1133-converse flexoelectric effects, piezoelectric effect, and electromagnetic stress are 1.64, 0.014, 0.30, and 0.29 nm, respectively. The displacement resulting from Joule heating is more than 10 orders of magnitude smaller than those from the other effects and can therefore be neglected. These results strongly support that the 1111-converse flexoelectric effect is the dominant mechanism driving device actuation, unless monolayer MoS₂ exhibits a μ_{1133} value that is significantly higher than μ_{1111} . Additionally, when these other effects are taken into consideration, μ_{1111} is estimated to be in the range of 1.41 ± 0.52 nC/m, under the assumption that $\mu_{1111} = \mu_{1133}$.

Table S1. Simulation parameters

Module	Parameter	Value
Solid Mechanics	Young's modulus, Au	70 GPa
	Poisson's ratio, Au	0.44
	Young's modulus, Ag	83 GPa
	Poisson's ratio, Ag	0.37
	Young's modulus, Parylene-C	2.75 GPa
	Poisson's ratio, Parylene-C	0.4
	Elastic modulus, C ₁₁ , MoS ₂	128.4 GPa ^[1]
	Elastic modulus, C ₂₂ , MoS ₂	128.4 GPa ^[1]
	Elastic modulus, C ₁₂ , MoS ₂	32.6 GPa ^[1]
	Elastic modulus, C ₃₃ , MoS ₂	38.3 GPa ^[2]
	Elastic modulus, C ₄₄ , MoS ₂	26.5 GPa ^[2]
	Elastic modulus, C ₅₅ , MoS ₂	26.5 GPa ^[2]
	Elastic modulus, C ₆₆ , MoS ₂	47.9 GPa ((C ₁₁ -C ₁₂)/2)
	Flexoelectric coefficient, μ_{1111} , MoS ₂	1.41 nC/m

	Flexoelectric coefficient, μ_{1133} , MoS ₂	1.41 nC/m
	Piezoelectric coefficient, e_{11} , MoS ₂	0.5 C/m ² [3]
	Isotropic structural loss factor (1/Q factor)	1/30
Electrostatics & Electric Currents	Terminal voltage, top electrode (Au)	20 V
	Terminal voltage, bottom electrode (Ag)	0 V
	Relative permittivity, Parylene-C	3
	Relative permittivity, MoS ₂	4 [4]
Heat Transfer in Solids	Thermal conductivity, Au	317 W/m·K
	Density, Au	19300 kg/m ³
	Heat capacity, Au	129 J/kg·K
	Thermal conductivity, Ag	429 W/m·K
	Density, Ag	10500 kg/m ³
	Heat capacity, Ag	235 J/kg·K
	Thermal conductivity, Parylene-C	0.084 W/m·K
	Density, Parylene-C	1289 kg/m ³
	Heat capacity, Parylene-C	712 J/kg·K
	Thermal conductivity, MoS ₂	23.2 W/m·K [5]
	Density, MoS ₂	5060 kg/m ³
	Heat capacity, MoS ₂	87.5 J/kg·K (14 J/mol·K) [6]
	Thermal Expansion (Multiphysics)	Thermal expansion coefficient, Au
Thermal expansion coefficient, Ag		1.89x10 ⁻⁵ /K
Thermal expansion coefficient, Parylene-C		3.5x10 ⁻⁵ /K
Thermal expansion coefficient, MoS ₂		7.6x10 ⁻⁶ /K [7]

Supplementary Figures:

Figure S7. (a) Resonant frequency trends for actuators with different MoS₂ layer thicknesses. A quadratically decreasing trend is maintained as the MoS₂ layer becomes thicker with increasing layer number. (b) Fifth-harmonic displacement for different MoS₂ layer thicknesses. Although the trend is maintained, the bi- and tri-layer MoS₂ actuators exhibit lower displacement compared to the monolayer MoS₂ actuator.

Figure S8. Fifth-harmonic displacement resulting from the 1111- and 1133-converse flexoelectric effects, piezoelectric effect, electromagnetic stress, and Joule heating. A 200 μm -long model with monolayer MoS_2 was implemented for the calculations, for which the 5th resonant frequency is estimated to be ~ 430 kHz.

Figure S9. Fifth-harmonic displacement of the monolayer MoS_2 actuator near the resonant frequency range for Q-factor calculation.

2. The actuation performance of monolayer and bilayer devices is similar. The deep reason should be reached. Is it purely flexoelectricity? If so, does the interface between layers or the slip plane play a role?

We appreciate the referee for this comment. While there is a possibility of slipping between the 2D layers, our calculations indicate that the lower actuation observed in the bilayer device compared to the monolayer device is mainly due to the increased flexural stiffness of the bilayer (vs. monolayer). We have added a brief explanation of this observation in the main text and included simulation results for thicker MoS_2 layers in the Supporting Information.

Revision to MS/SI

In main text, page 9-10:

To study the effects of MoS_2 thickness on the actuation behavior, we simulated displacement and resonance frequencies of bi- and tri-layer MoS_2 . The resonant frequency and displacement trends remained consistent for bi- and tri-layer MoS_2 actuators (Figure S7a). In addition, increasing the MoS_2 thickness from monolayer to bilayer reduces the harmonic

displacement of the actuator, which aligns with our experimental observation that the bilayer device exhibits approximately 18% lower displacement than the monolayer counterpart. This reduction is primarily attributed to the increased flexural stiffness resulting from the additional MoS₂ layer.

In Supporting Information:

Figure S7. (a) Resonant frequency trends for actuators with different MoS₂ layer thicknesses. A quadratically decreasing trend is maintained as the MoS₂ layer becomes thicker with increasing layer number. (b) Fifth-harmonic displacement for different MoS₂ layer thicknesses. Although the trend is maintained, the bi- and tri-layer MoS₂ actuators exhibit lower displacement compared to the monolayer MoS₂ actuator.

3. The actuator demonstrates impressive performance at its resonant frequency (~19 kHz). However, for many practical applications, a broader frequency band is highly desirable. Could the authors comment on the expected performance of the device when operated off-resonance? What potential strategies could be explored to achieve a wider bandwidth?

We thank the referee for this valuable insight. The off-resonance displacement is ~5 nm, 1/10 of the resonant displacement. Although our actuator has a characteristic operation frequency, the resonant frequency and the width of the resonant peak of the actuator can be tuned by adjusting the length, thickness, elastic modulus, and viscosity of the supporting layer. Furthermore, maximizing the number of electrode combs would enhance both off-resonant and resonant actuations. To reflect the referee's comments, a discussion of these considerations has been added to the manuscript.

Revision to MS/SI

In main text, page 10:

As our measurements of the actuator displacement were acquired at the 5th resonant frequency, there is a significant potential to further enhance the device performance by utilizing lower-order harmonic resonance. Reducing the width and pitch of the electrode combs to maximize the number of combs in the device would further improve actuation performance. Furthermore, the resonant frequency and the width of the resonant peak of the actuator can be tuned by adjusting the length, thickness, elastic modulus, and viscosity of the supporting layer to meet specific application requirements, offering a wide range of applicability.